# Effect of Oregano Essential Oil on Growth and Composition of Gut Prokaryote Microbiota on Striped Bass (*Morone saxatilis*)

**DOI:** 10.3390/microorganisms13020264

**Published:** 2025-01-25

**Authors:** Dora Alejandra Trejo-Ramos, César Omar Rodríguez-Arana, Roberto Cruz-Flores, Mónica Hernández-Rodríguez

**Affiliations:** 1Centro de Investigación Científica y Educación Superior de Ensenada (CICESE), Carretera Ensenada-Tijuana No. 3918, Zona Playitas, Ensenada 22860, Baja California, Mexico; dtrejo@cicese.edu.mx; 2Pacifico Aquaculture S.A.P.I. de C.V., Km 104 Carretera Tecate-Ensenada, Esq. Calzada Vista al Mar, El Sauzal de Rodriguez 22760, Baja California, Mexico; rasecramo@hotmail.com

**Keywords:** *16s rRNA*, amplicon sequencing, oregano essential oil, striped bass, prokaryote gut microbiota

## Abstract

Sustainable strategies, such as oregano essential oil (OEO), are being assessed to improve farmed fish’s health and performance. Several studies in freshwater species describe the beneficial effects of OEO as a dietary supplement. Nonetheless, information about its use in marine fish is scarce. Hereby, this study consisted of four experimental groups: a control and three levels of OEO dietary inclusion (OEO75 (0.75 mL/kg), OEO100 (1.0 mL/kg), and OEO125 (1.25 mL/kg)) with 23 fish of 110 ± 1.2 g per tank (*n* = 276) in a seawater flow-through system. After 70 days, data for growth parameters and samples for gut microbiota were taken. The final weight of OEO75 was higher (242.3 ± 24.2 g), and its feed conversion rate (0.91 ± 0.0) was reduced. However, these differences could be related to the sampling interval between the groups. Bioinformatic analysis of OEO groups revealed a reduction in *Proteobacteria* (*Vibrio*, *Flavobacteria*, and *Vibrionales* order) abundance and an increase in *Euryarchaeota* phyla in a dose-dependent manner. The predicted functions shifted from nutritional (OEO75) to replication, proliferation, and bacterial defense mechanisms (OEO100 and OEO125). These results show that adding OEO did not improve growth performance, but it reshaped the gut microbiota by reducing the abundance of dominant bacteria and modifying their metabolic pathways.

## 1. Introduction

The exponential increase and ongoing expansion of aquaculture production stand out as a solution to provide more nutritious aquatic products to ensure food security for the ever-growing global population [1]. The escalating international demand for fish supply has promoted significant advancements in the aquaculture sector, leading to the widespread adoption of intensive farming systems. However, intensive production practices can lead to stressful conditions for finfish, such as high stocking densities, changes in water quality, and handling, compromising fish’s welfare and health [2,3]. The aquaculture sector is exploring new sustainable strategies, such as phytogenics, to improve the growth and health of farmed fish by ensuring intestinal health and balance of gut microbiota [4,5].

The intestine is a crucial organ for optimizing fish performance, owing to its role in nutrient digestion and absorption and its active involvement as a primary defense barrier alongside the microbiota [6,7]. This gut microbiota is complex as it comprises protozoa, yeast, viruses, bacteria, and archaea [8]. Studies have shown a relationship between the gut microbiota and the general health of fish, recognizing it as an extra “organ” due to its significant effect on host growth and physiology functions [9,10,11]. Among the mechanisms elucidated for gut bacterial species, their contribution lies in their ability to influence host growth by modulating feeding behavior by producing bioactive compounds and metabolites [12,13]. These bacteria support digestion by secreting glucosidases, amylases, proteases, cellulases, and lipases [14,15]. Additionally, gut bacterial communities actively engage in the host’s metabolic processes, promoting the upregulation of functional pathways associated with the host’s current microbiota [16]. Moreover, studies have demonstrated gut microbiota’s facilitative role in enhancing essential fatty acid uptake [17]. In the archaea domain, little information is known about its functions or effects as members of fish’s microbiota; however, some of the phylotypes detected by genomic approaches are *Euryarchaeota*, *Thaumarchaeota Nanoarchaeota*, *Korarchaeota,* or unidentified archaea [8]. Recently, archaea have gained attention as potential probiotics due to their diverse beneficial qualities. Firstly, they can resist the routinary mechanical and thermic processes of animal feed production, thereby remaining viable to colonize the gut and prevent the invasion of pathogens [18,19]. To date, no pathogenic archaea have been identified. On the contrary, they are integral components of the microbiota associated with animals and humans, with the ability to modulate the host’s immune system [19,20].

Phytogenics are products obtained from plants used as functional feed additives in animal production due to their antioxidant and anti-inflammatory properties [21]. These products encompass various plant components, including leaves, roots, flowers, seeds, or concentrated extracts rich in active compounds such as essential oils [22,23]. OEO is described as an efficient additive for the fish diet [3]. Studies have documented the various properties of OEO, including its roles as an immunoregulator, antioxidant, enhancer of gut morphology, stimulator of digestive enzymes, and modulator of microbiota [24,25,26,27]. Carvacrol and thymol, the predominant phenolic components of oregano oil, are closely associated with their diverse physiological effects [4,24]. Most studies of OEO have centered on continental species such as the common carp (*Cyprinus carpio*) and Nile tilapia (*Oreochromis niloticus*) [25,28,29,30,31]. Recently, some studies on marine fish have reported the use and effects of OEO [3]. In turbot (*Scophthalmus maximus*) [27], the administration of 1 mL/kg dose reduced the relative abundance of pathogenic bacteria such as *Listonella* and *Sphingomonas*, while concurrently promoting the proliferation of beneficial bacteria associated with positive effects on gut health. Also, on the hybrid grouper (*Epinephelus fuscoguttatus* ♀ × *Epinephelus lanceolatus* ♂) [32], an addition of 0.3 g/kg of OEO promoted the increase in beneficial bacteria and decreased pathogenic bacteria like *Vibrio* spp.

The striped bass (*Morone saxatilis*) is an anadromous, endemic fish of the North American Atlantic Coast and the Gulf of Mexico. Following their introduction to the West Coast, they are now distributed from British Columbia, Canada, to Baja California, México [33]. Their potential for aquaculture is highlighted by their euryhaline nature. In addition, knowledge of their biological and rearing requirements is available due to the development of the hybrid striped bass (*Morone chrysops* × *Morone saxatilis*) culture [34]. Currently, Pacifico Aquaculture, located in Ensenada, México, is the only commercial farm that produces striped bass raised in sea cage culture.

By investigating the effects of OEO supplementation in the context of aquaculture, we aim to contribute knowledge on its possible inclusion as a sustainable dietary additive to optimize fish health and productivity. Considering the relationship between gut health, microbiota composition, fish growth performance, and the imperative need for new sustainable alternatives, this study evaluated the effect of diets supplemented with oregano oil on growth performance and gut bacterial and archaeal microbiota composition of striped bass juveniles raised in seawater in Ensenada, Baja California, México.

## 2. Materials and Methods

The bioassay was conducted at the facilities of the Aquaculture Department of the Ensenada Center for Scientific Research and Higher Education (CICESE, for its acronym in Spanish) located in Ensenada, Baja California, México.

All the procedures performed in this study for the experimental culture of striped bass followed the guidelines in The Care and Use of Fish in Research, Teaching, and Testing by the Canadian Council on Animal Care (CCAC) and the American Veterinary Medical Association (AVMA) Guidelines for the Euthanasia of Animals [35,36].

### 2.1. Fish and Culture Conditions

Striped bass (*M. saxatilis*) juveniles were donated by Pacifico Aquaculture SAP.I de C.V. located in Ensenada, Baja California, México. Before bioassay, fish had an acclimation period of one month in the experimental system, which consisted of a seawater flow-through system composed of 12 tanks of 430 L. Additionally, each tank had a temperature controller, a titanium heater of 1000 watts, and a temperature and water sensor. Water parameters were maintained as follows: water temperature (23.0 ± 0.04 °C), salinity (33.9 ± 0.5‰), dissolved oxygen (6.5 ± 0.3 mg/L), TAN (<0.5 mg/L), nitrite (<0.25 mg/L) and nitrate (<5.0 mg/L). During this period, animals were fed with the commercial diet EWOS^®^ (Cargill, Vancouver, BA, Canada) comprising 54% crude protein, 14% crude fat, 10% nitrogen-free extractives, 0.6% crude fiber, 1.5% phosphorus, 0.6% sodium, 11% ash, and 10% moisture, according to the manufacturer’s guaranteed analysis.

At the beginning of the bioassay, juveniles maintained in the experimental system had an initial weight of 110 ± 1.2 g and a total longitude of 21 ± 0.1 cm. Twenty-three fish (*n* = 276) were in each tank with a water flow of 1.50 L/min (water changes of five times the tank’s volume). Water temperature (22.9 ± 0.01 °C), salinity (34.7 ± 0.03‰), and dissolved oxygen (6.5 ± 0.07 mg/L) were recorded daily with a multiparameter (YSI, Yellow Spring, OH, USA). While TAN (<0.5 mg/L), nitrite (<0.25 mg/L), and nitrate (<5.0 mg/L) were evaluated every three days with a commercial kit (API^®^, Chalfond, PA, USA).

### 2.2. Experimental Diets and OEO

The oregano essential oil (OEO) at 5% is a commercial product (Orego-Stim^®^) of Anpario plc. (Nottinghamshire, UK) which contains essential oil extracted from *Origanum vulgare* subsp. *hirtum*. The foundational feed used was the commercial diet EWOS^®^ (Cargill, Vancouver, BA, Canada). The OEO, diluted to a 2.5% concentration with distilled water, was applied to this base feed via spraying [27]. The OEO dose was divided into two sprinkles, and the diet was mixed in a blender (Member’s Mark, Bentonville, AR, USA) with plastic blades to ensure the pellets were evenly impregnated. The feed was dried in an incubator (Boekel Scientific, Philadelphia, PA, USA) at 21 ± 1.0 °C for 20 min; these conditions were standardized in preliminary tests before the bioassay. To ensure the stability of the OEO, the dried diet was prepared 2 h before feeding and stored in plastic containers at 4 °C [37] both before and during feeding.

### 2.3. Experimental Design

In this experiment, four groups were designed: a control group (0 mg/L) and three treatments with OEO (OEO75 (0.75 mL/kg), OEO100 (1.0 mL/kg), and OEO125 (1.25 mL/kg). These concentrations were determined based on the Pacifico Aquaculture company’s preferences and following the manufacturer’s recommendations. The fish were fed twice a day to apparent satiety with the experimental diets for 70 days.

### 2.4. Growth Performance and Feed Utilization Parameters

Initial biometry of all fish in each tank was conducted, recording both weight (g) and length (cm). Growth measurements were taken every 14 days until the end of the experiment. The hepatosomatic index was calculated at the beginning and the conclusion of the bioassay. Growth performance and feed utilization parameters were evaluated in terms of [38,39,40]:Survival (S, %) = 100 × (final number of fish/initial number of fish)(1)Weight gain (WG, g) = final fish body weight (FFBW)−initial fish body weight (IFBW)(2)Weight gain (WG, %) = 100 × [(FFBW−IFBW)/IFBW](3)Specific growth rate (SGR, %/day = 100 × [(Ln FFBW−Ln IFBW)/days of bioassay](4)Feed conversion ratio (FCR) = feed intake/WG(5)Hepatosomatic index (HSI, %) = 100 × (Liver weight/fish body weight (FBW))(6)Allometric condition factor (K_a_) = 100 × (FBW/total fish length ^b^)(7)

### 2.5. Sampling Collection and DNA Extraction

Before starting the experiment, an initial sample (*n* = 10) was collected of striped bass juveniles (79 ± 17.4 g) reared in a seawater-recirculating aquaculture system (*n* = 5) and a flow-through system (*n* = 5) at 23 °C and feed with EWOS^®^ commercial diet for the microbiome analysis. To establish this as our initial sample, we verified that no differences were detected in the alpha and beta diversity metrics (see Appendix A). All fish remained without food for 14 h before the final sampling. Five fish per tank were euthanized using a two-step procedure. Initially, fish were anesthetized using 0.05 mL/L clove oil (Sigma-Aldrich, St. Louis, MO, USA) until sedation. Following anesthesia, pithing was performed to ensure rapid euthanasia, adhering to established ethical standards for fish welfare and scientific research [36]. The distal intestine was aseptically dissected and kept at −80 °C in an ultra-freezer (Thermo Fisher Scientific, Waltham, MA, USA) until DNA extraction. A total of 49 samples (including the initial samples) of the distal intestine were used to extract total DNA using DNAeasy^®^ Blood and Tissue (Qiagen, Hilden, Germany). The quality and concentration of total DNA (ng/µL) were measured with a NanoDrop 2000 Spectrophotometer (Thermo Fisher Scientific). The isolated DNA was sent to Omega Bioservices, Norcross, GA, USA, for Next Generation Sequencing.

### 2.6. Library Preparation

Before library construction, dsDNA concentration was measured with QuantiFluor^®^ dsDNA System (Promega, Madison, WI, USA). Libraries were prepared using the kit KAPA HyperPlus (Roche, Basel, Switzerland), amplified with *16S rRNA* primers IlluminaF (5′-CCTACGGGNGGCWGCAG-3′) and IlluminaR (5′-GACTACHVGGGTATCTAATCC-3′) targeting *16S rRNA* V3–V4 regions and sequenced on an illumina^®^ MiSeq platform (Illumina, Inc., San Diego, CA, USA) using a 2 × 300 bp paired-end protocol. These procedures were carried out by Omega Bioservices, Norcross, GA, USA.

### 2.7. Bioinformatic Analysis

The raw sequence data were processed in Geneious Prime version 2024.0.3 [41]. Paired-end reads were joined and filtered to remove adapters and low-quality bases (below Phred quality score of 30) on both ends of the read with the BBDuk plug-in (Geneious Prime^®^). After reads were merged using BBmerge (version 2024.03), reads associated with the host were removed [42]. To eliminate host-related reads, the mitochondrial genome of *M. saxatilis* was used as the reference for mapping all sequences, with a particular focus on reads obtained from the initial samples. Additionally, only sequences ranging from 400 to 480 base pairs were selected for bioinformatic analysis, ensuring a targeted approach to data interpretation. Sequences, sample metadata, and manifest files were uploaded to the EasyMAP (http://easymap.cgm.ntu.edu.tw, accessed on 11 March 2024) portal [43].

Using the QIIME2 (through EasyMAP-http://easymap.cgm.ntu.edu.tw, accessed on 11 March 2024) pipeline, operational taxonomic units (OTUs) were picked by clustering reads through a *de novo* assembly. Taxonomy assignment was performed using the Greengenes V3–V4 classifier. For the OTUs table and taxonomy analysis, a 97% confidence for sequence similarity was considered. Diversity metrics were calculated for microbiota OTUs for alpha diversity. Observed OTUs and the Shannon index (species richness in the groups) were used, and statistical differences were analyzed using a Kruskal–Wallis test with a significance of *p* < 0.05. Beta diversity (effect of OEO on microbiota composition between groups) was assessed using permutational multivariate analysis of variance (PERMANOVA) based on the distances of weighted UniFrac (phylogeny and abundance) [44], and principal coordinate analysis (PCoA) was used to observe the similarity/dissimilarity between the samples [45].

### 2.8. Taxonomy Differential Abundance and Function Prediction

An extended analysis of taxonomy differential abundance (TDA) was performed on the EasyMap platform using the linear discriminant analysis effect size (LEFSe) [46], plug-in applying the KEGG database (through EasyMAP-http://easymap.cgm.ntu.edu.tw, accessed on 11 March 2024) at the 6 level and the Kruskal–Wallis test with a significance of *p* < 0.05. The logarithmic Linear Discriminant Analysis (LDA) score set value was 2 [43]. Using the results of TDA, the microbiota function was predicted through the PICRUSt (through EasyMAP-http://easymap.cgm.ntu.edu.tw, accessed on 11 March 2024) by mapping the Greengenes ID to recognize the corresponding function in the KEGG database at level 3 [47]. The predicted microbiota functions were visually represented using a bar graph based on LDA scores (log10).

### 2.9. Statistics

Data of growth and feed utilization parameters are presented as mean ± standard deviation (SD) of three replicates. Before statistical analysis, all data were tested for normality and variance homogeneity using the Shapiro–Wilk and Bartlett tests, respectively. Data were analyzed by one-way analysis of variance (ANOVA), and when overall differences were significant (*p* > 0.05), Tuckey’s test was used to compare the mean values between individual treatments. A Kruskal–Wallis nonparametric test and Dunn’s post hoc were applied when assumptions were unmet. All statistical analyses were performed with an alpha level of 0.05 using Statistica version 10.0 software (StatSoft Inc., Tulsa, OK, USA).

## 3. Results

### 3.1. Growth Performance and Feed Utilization

The final body weight (FBW) of fish from the OEO75 treatment was higher than the control group and OEO100 (*p* < 0.05) (Table 1). The fish’s feed conversion rate decreased from the OEO75 treatment (*p* < 0.05) compared with the control, OEO100, and OEO125. A decreasing trend on the HSI was observed as the OEO dose increased. Nevertheless, non-statistical differences were found in the other growth and productive parameters (WG (g), WG (%), SGR, HSI, K_a_). The survival rate of the juveniles was 100% in all groups at the end of the experiment.

### 3.2. Gut Microbiota Taxonomic Composition Analysis

The OTUs table construction of bacteria taxonomy assignment was carried out using a total of 1,056,160 sequence reads (Q > 30), where a total of 4394 representative sequences were identified and used as a reference to cluster reads into OTUs. The relative abundance of bacteria (71–94%) and archaea (6–28%) were recognized at the domain level. Significant differences at the domain level were observed between the initial samples and the experimental groups. The archaea domain was more abundant in the experimental reads (28%) compared to the initial sample reads (6%). Conversely, the bacteria domain was more prevalent in the initial samples (94%), as illustrated in Figure 1.

A total of 22 (1 unclassified) bacteria and 2 archaea phyla were identified and assorted in 3463 (454 unclassified) genera. The most representative bacteria phyla were *Proteobacteria* (68–43%), *Firmicutes* (8–10%), and *Bacteroidetes* (0–5%). Archaea phyla were composed of *Euryarchaeota* (9–38%) and others (<1.5%). The initial samples main bacteria and archaea genera were represented by *Methylobacterium* (22%), *Escherichia* (16%), *Bradyrhizobium* (6%), unclassified bacteria (4%), and *Methanosaeta* (3%). A noteworthy trend emerged in the relative abundance of the *Proteobacteria* phylum, which exhibited a marked decrease with increasing doses of OEO. Additionally, the *Bacteroidetes* phylum was less present in OEO75 (0.26%) than in the other samples (Figure 2). Differences were also recorded at the genus level, where the main bacteria and archaea genera found in the experimental groups were *Photobacterium* (8–31%), *Vibrio* (6–22%), *Methanosaeta* (8–13%), *Methanobacterium* (8–15%), and *Methanobrevibacter* (3–6%). Curiously, the *Vibrio* genera were less abundant in the initial samples (2%), OEO125 group (6%), and OEO75 group (9.62%). Consequently, the *Vibrionales* order showed low relative abundance in the OEO75 (3.5%) and the OEO125 group (2.2%) (Figure 3).

### 3.3. Gut Microbiota Richness and Diversity

Alpha diversity, the Shannon diversity index and observed OTUs of the gut microbiota in the experimental groups (control, OEO75, OEO100, and OEO125) were significantly higher than those in the initial samples (*p* < 0.01). Although no statistical differences were found between the experimental groups themselves (*p* > 0.05), the control group exhibited notably lower variability in richness compared to the OEO groups (Figure 4).

For the beta diversity, PCoA separately clustered the initial samples and experimental groups, forming two groups (Figure 5). These differences between experimental groups were further corroborated by PERMANOVA using the weighted UniFrac distances, where a dissimilarity was evidenced in all paired comparisons between the initial samples and experimental groups (control, OEO75, OEO100, and OEO125) (*p* < 0.01).

### 3.4. Differential Gut Microbiota Abundance in Juveniles Fed with OEO-Supplemented Diets

The extended TDA revealed differences in the abundance of taxa between initial samples, control group, OEO75, and OEO100 (*p* < 0.05) (Figure 6A). The control group exhibited the highest abundance of taxa (12 taxa) compared to the initial samples (5 taxa), OEO75 (2 taxa), and OEO100 (1 taxon) groups. The archaea and bacteria taxa associated with the control group were class: *Thaumarchaeota* and *Flavobacteriia*; order: *Vibrionales*, *Cernarchaeales*, *Oceanospirillales*, and *Rhodobacterales*; family: *Corynebacterium*; genera: *Vibrio*, *Corynebacterium*, *Nitrospopumilus*, and *Methylobacterium* (*p* < 0.05). These taxa differed (*p* < 0.05) from the initial sample’s family: *Caulobacteraceae*; genera: *Paracoccus* and *Sediminicola*; the OEO100 group genera: *Photobacterium*; and the OEO75 group family: *Pelagibacteraceae* and genera: *Rheinheimera*. Additionally, some of these differentially expressed taxa were phylogenetically different between the groups (*p* < 0.05) (Figure 6B).

### 3.5. Identification of Microbiota Functionality

The incorporation of PICRUSt with the KEGG database to the results of LEfSe showed differences in the predicted functions of the microbiota of the initial samples and experimental groups (*p* < 0.05). In decreasing order, the most enriched group was the initial samples with 58 predicted functions, followed by the OEO100 group with 12 predicted functions, then the OEO75 and OEO125 groups with seven predicted functions, respectively, and the control group with six predicted functions (Figure 7, and Appendix A for the complete figure).

The microbiota function of initial samples was mainly related to the metabolism of macronutrients (carbohydrates, amino acids, fatty acids), biodegradation and metabolism of xenobiotics (toluene, styrene, and naphthalene degradation), diseases (viral and parasitic), and bacterial growth and death (cell cycle, meiosis, apoptosis) (*p* < 0.05). The control enriched pathways for experimental groups were mainly related to nutrient acquisition (carbohydrate and amino acid metabolism, phosphotransferase system) and partially to bacterial transcription factors. In contrast, a clear trend was observed in the groups receiving the experimental diets as the dose of OEO increased. Initially, functional pathways were associated with nutritional roles such as carbohydrate, amino acid, vitamin, and cofactor metabolism (OEO75). As the dosage increased (OEO100 and OEO125), pathways shifted towards bacterial replication (DNA replication proteins, translation factors, nucleotide metabolism), bacterial proliferation (e.g., *Vibrio cholerae* infection, epithelial cell signaling in *Helicobacter pylori* infection), and bacterial defense mechanisms (restriction enzymes, sporulation) (*p* < 0.05).

## 4. Discussion

With the ongoing growth of the aquaculture industry, new strategies such as phytogenics are being employed to promote fish productivity as a sustainable approach to mitigate stress effects on growth, health, welfare, and the consequent impact on production [48]. Therefore, medicinal plants are presented as a promising, safer, and cheaper alternative for preventing and controlling fish diseases in aquaculture [49]. In the search for medicinal plants with promising bioactive compounds, essential oils of aromatic plants have been studied, with OEO being the most frequently used for its properties on animal performance [3]. However, it is essential to highlight that its effects on the gut microbiota of marine fish and its potential impact on fish growth remain limited. Clarifying this relationship could provide valuable insights into the potential of OEO as a possible solution to enhance fish performance in intensive farming systems.

Several studies have documented the effects of OEO as a growth enhancer by boosting final weight, weight gain, feed conversion ratio, condition index, and protein efficiency ratio [3,31,50,51]. Some of the properties attributed to OEO for this enhancement are its capacity to stimulate digestive enzyme activity [26], improve intestinal morphometry, and, hence, absorption efficiency [25,52]. In this study, the striped bass from the OEO75 treatment had the highest FW and lowest FCR of all the experimental groups. However, during the 70 days of the bioassay, no differences were observed in the fish’s productive parameters (average weight, WG, and SGR) in the different treatments. Thus, the higher final FW and reduced FCR of the fish of OEO75 can be attributed to the timing of sampling across groups, with a 7-day interval between the initial (control) and final (OEO75) samples. Nevertheless, while the effect of OEO as a growth promoter remains noteworthy, it may be more pronounced in striped bass during the early juvenile stage, as observed in other studies where significant improvements were reported in small yellowtail tetra (*Astyanax altiparanae*) [53]; rainbow trout (*Oncorhynchus mykiss*) [50]; koi carp (*Cyprinus Carpio*) [26]; hybrid grouper [32]. Furthermore, it is essential to highlight that the differences between studies could depend on factors related to animal culture (fish age, environment conditions, stocking density) and the OEO itself (the composition and level of inclusion of OEO) [54,55].

A decrease in HSI was observed in fish in the experimental groups as the OEO level increased; however, there were no differences between treatments. This result agrees with the reported in channel fish (*Ictalurus punctatus*) fed with diets containing the same OEO product used in our study, these fish had a lower HIS. On the contrary, the fish that received other experimental diets (Thymol, Carvacrol, and Carvacrol + Thymol) showed a higher HSI [51]. The authors mentioned that other trace active compounds of this OEO product were responsible for this effect or could even act synergically with the main compounds of OEO (carvacrol and thymol) [22,51].

For decades, growth promoters of antibiotic origin were dominant and commonly used to improve animal productivity, as their chemical qualities might allow them to act on intestinal microbiota. However, the actual situation of microorganism resistance and public health made indispensable the search for new functional compounds with characteristics to modulate directly or indirectly gut microbiota as a key to influencing animal performance [51,56].

It is well-established that the intricate relationship between a fish host and its microbiota is influenced by a range of host-specific factors, including age, immune response, genetics, sex, trophic level, and diet, as well as environmental factors such as temperature and salinity [9,57,58,59]. To reference the composition of striped bass’s microbiota, we took an initial sample with conditions similar to salinity, diet, and temperature, as the experimental system did before OEO was added. However, our results reveal striking differences across multiple levels—from domain to genus—showcasing varying richness, phylogenetic diversity (dissimilarity), differential taxonomic abundance, and predicted metabolic pathways within the microbiota. These results underscore the significance of the rearing system and the organism’s developmental stage as crucial factors to consider. The water composition in the culture system depends on the organic matter and the provision of a microbial load for the system and the water source [60]. In this context, the continuous exchange of water in the flow-through system decreases organic nutrients and bacterial density, while in recirculation systems, it is maintained due to the low water exchange, promoting the proliferation of slow-growing microbes [60,61].

Furthermore, it has been demonstrated that as the organism grows, the diversity of its bacterial microbiota increases [9], possibly due to the expansion of epithelial surfaces, which provides a larger area for bacterial colonization. The anterior aligns with our initial observations, where younger organisms in the initial sample exhibited similar bacterial diversity regardless of rearing conditions. This underscores the intricate interactions between host-specific and environmental factors in shaping the host microbiota, which may contribute to the observed divergence between the initial samples and the experimental groups.

Another aspect attributed to OEO as a growth promoter is its capacity to modulate fish microbiota either by its antibacterial activity or immunomodulation [37]. Numerous studies have demonstrated the antibacterial properties of OEO, highlighting its notable lipophilic characteristics. This property of OEO damages the integrity of the cell membrane, altering its stability and causing cell death [5,62,63,64]. Additionally, the powerful antioxidant ability to reduce oxidant molecules or to delocalize an unpaired electron in their phenolic aromatic ring protects cells against oxidation [55,65]. These actions of OEO likely improve gut health and mucosal integrity by defending against oxidative damage, reducing inflammation, and promoting an appropriate immune response to pathogens [5,37,66,67].

Our results confirm a significant effect of OEO on the composition of the gut prokaryote microbiota of striped bass, primarily through modulation of bacterial community and its predicted functions. This conclusion is supported by several findings, notably the marked decrease in *Proteobacteria* (*Photobacterium*, *Vibrio*, and *Vibrionales* order) and simultaneously the increase in *Euryarchaeota* (*Methaobacterium*, *Methanobrevibacter*, *Metahocorpusculum* and, *Methaosaeta*) in a dose-dependent manner.

Archaea, although typically less abundant than bacterial communities in fish gut ecosystems, possess unique adaptations such as cell wall structures that enable survival in extreme conditions, along with distinctive metabolic processes like methanogenesis [68,69]. Methanogens from *Euryarchaeota* play a crucial role in the final stages of carbohydrate metabolism, converting organic matter by-products into methane [69,70]. On the other hand, the antibacterial effects of OEO have been predominantly observed against Gram-positive and Gram-negative bacteria, with limited information on its impact on archaea [62,71]. However, studies have demonstrated that combinations of essential oils from cinnamon, oregano (*Thymus capitatus* L.), rosemary, eucalyptus, and dill seeds can reduce rumen methanogenic archaea in vitro [72]. In contrast, in vivo studies supplementing lamb with cinnamaldehyde, garlic oil, and juniper berry oil have shown an increased abundance of methanogenic archaea like *Methanobrevibacter smithii* and *Methanospaera stadtmanae* potentially mediated through effects on rumen protozoa community [73]. Whether similar microbial interactions occur in the fish gut remains unclear, raising the possibility that archaeal methanogens in striped bass may exhibit greater resilience to OEO, leaving a new approach to address in future research.

Despite expectations that the antibacterial properties of OEO would reduce species variability in the OEO groups, our study found that the control group exhibited lower variability in richness and observed OTUs. Moreover, the control group showed a higher differential abundance of the *Flavobacteriia* class, *Vibrio*, and *Vibrionales* order.

*Vibrio* and *Flavobacteria* species are recognized as opportunistic pathogens in marine species, including striped bass, often triggering infections following stress events [74,75]. Furthermore, these bacteria are prevalent in the gut of marine fish not only due to their evolved relationships as potential symbiotic, mutualistic, or commensal microbiota but also because they have diverse, efficient virulence mechanisms to colonize the intestinal environment [76,77]. For instance, *Vibrio* can physically attach to the intestinal mucus layer, competing with other microorganisms by secreting compounds to prevent their growth, proliferating, and evading host defenses [78]. As members of the *Bacteroidota* phyla, *Flavobacteria* have been described as having similar virulence factors that inhibit other bacteria, colonize, and proliferate on their host [79]. These bacteria could employ all these traits as “dual-use”, serving pathogenic processes and as an adaptative strategy to grow and survive in the host and environment [77,80]. Hence, it is plausible that the lower variability of richness observed in the control group could be attributed to the higher dominance of *Vibrio* and *Flavobacteria*. This aligns with the differential abundance of these bacteria in the control group, the decrease in the Proteobacteria phyla and *Vibrionales* order and *Vibrio* genera in the OEO groups, and at the same time, the increase in the variability of richness and observed OTUs in these groups.

The effect of OEO as an immunostimulant is due to its antioxidant characteristic, which contributes to regulating the excessive growth of bacteria. Studies in channel catfish, koi carp, rainbow trout, and gilthead seabream (*Sparus aurata* L.) have reported the immunomodulation of oregano by improving innate immunity, enhancing the activity of lysozyme, complement activity, and levels of immunoglobulin M. Additionally, promoting cellular immunity by increasing lymphocytes number and phagocytic activity of leucocytes [26,51,81,82]. Hence, this characteristic of OEO may play a pivotal role in shaping bacterial communities, as evidenced in this study. In this sense, future studies could additionally employ genomic and transcriptomic approaches to understand the effects of OEO at molecular levels. Interestingly, within the OEO-treated groups, bacterial functional pathways transitioned from metabolic activities focused on nutrition and digestion (macromolecule metabolism) to processes involving replication, proliferation, and defense mechanisms (such as sporulation and restriction enzymes). For instance, the ability of bacteria to switch from a vegetative state to sporulation allows them to cope with adverse growth conditions, employing this mechanism only when active growth is untenable [83]. In addition, restriction enzymes are used as a defensive system to safeguard bacteria when viruses or bacteriophages invade them [84]. These changes in the predicted metabolic pathways of fish intestinal microbiota show an insight into the microbiota functions [85] and may indicate the effects of OEO on bacterial metabolism. Therefore, this finding suggests that the dual properties of OEO—its antibacterial efficacy and immunostimulatory impact—may have prompted bacteria to shift from nutritional activities to a defensive role.

The *16S rRNA* amplicon sequencing technology allowed us to identify the archaeal and bacterial taxa of striped bass’s gut microbiota and its variations between the experimental groups. However, it is important to acknowledge that a limitation of *16S rRNA* amplicon sequencing analysis is its inability to distinguish between environmental DNA and DNA from viable cells. Unlike most RNA species, which quickly degrade, DNA remains detectable after a cell’s death [86]. Therefore, a strategy to determine living taxa is using RNA to obtain complementary DNA and then applying the *16S rRNA* amplicon sequencing analysis [87]. However, cost and differences in processing steps should be considered. Although the RNA approach would have increased analysis sensitivity, the acquired data allowed for a comprehensive analysis. These findings are valuable for innovating and applying new sustainable alternatives based on aromatic plants like oregano.

## 5. Conclusions

Overall, our results show that the addition of OEO did not directly affect growth performance. However, it did reshape striped bass’s gut microbiota by reducing the relative and differential abundance of dominant opportunistic pathogens such as *Vibrio* and *Flavobacteria* and allowing other potential benefit microorganisms such as archaea to increase their abundance. Additionally, the supplementation with OEO at doses over 1.0 mL/kg modifies the metabolic pathways of the microbiota from a nutritional role to a defensive role, as it could have been an effect of the exposure to OEO antibacterial and immunostimulant properties reflected in their metabolic profiles. Finally, it would be interesting to explore if these changes in the gut microbiota profile could impact fish growth on a long-term basis.

## Figures and Tables

**Figure 1 microorganisms-13-00264-f001:**
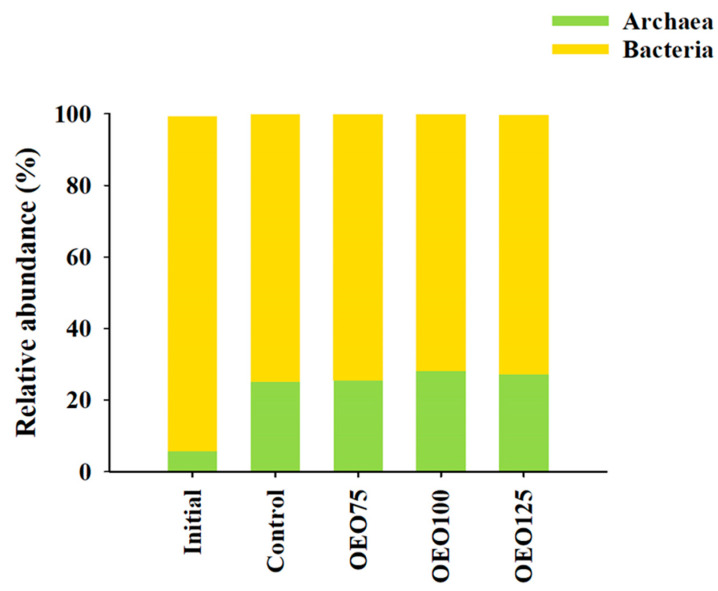
Taxonomic composition at the domain level of initial samples and experimental groups (Control, OEO75, OEO100, and OEO125) of striped bass (*M. saxatilis*) archaeal and bacterial gut microbiota. Observe that the archaea domain is less abundant in the initial samples than in the experimental groups.

**Figure 2 microorganisms-13-00264-f002:**
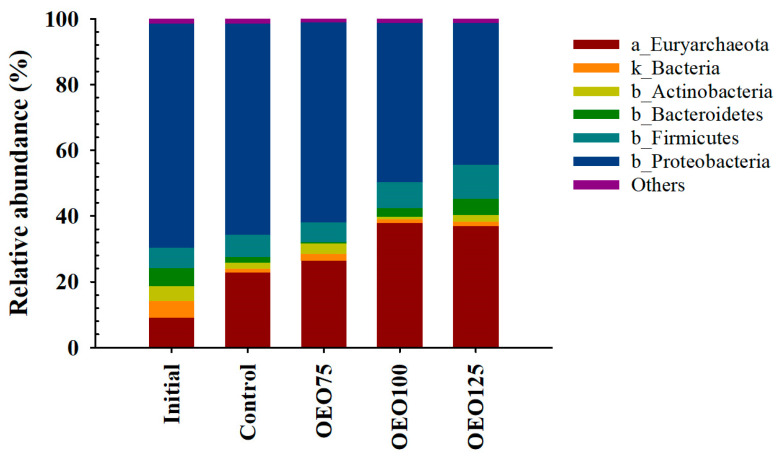
Taxonomic composition at phylum level of initial samples and experimental groups (Control, OEO75, OEO100, and OEO125) of striped bass (*M. saxatilis*) archaeal (a) and bacterial (b) gut microbiota. Some bacterial reads were only classified to domain level (k).

**Figure 3 microorganisms-13-00264-f003:**
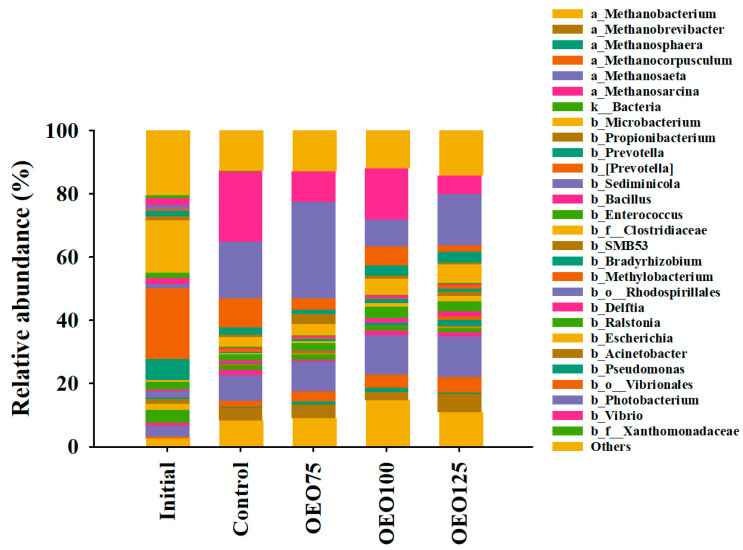
Taxonomic composition at genus level of initial samples and experimental groups (Control, OEO75, OEO100, and OEO125) of striped bass (*M. saxatilis*) bacterial (b) and archaeal (a) gut microbiota. Some bacterial reads were only classified to domain level (k_), order level (b_o), and family level (b_f).

**Figure 4 microorganisms-13-00264-f004:**
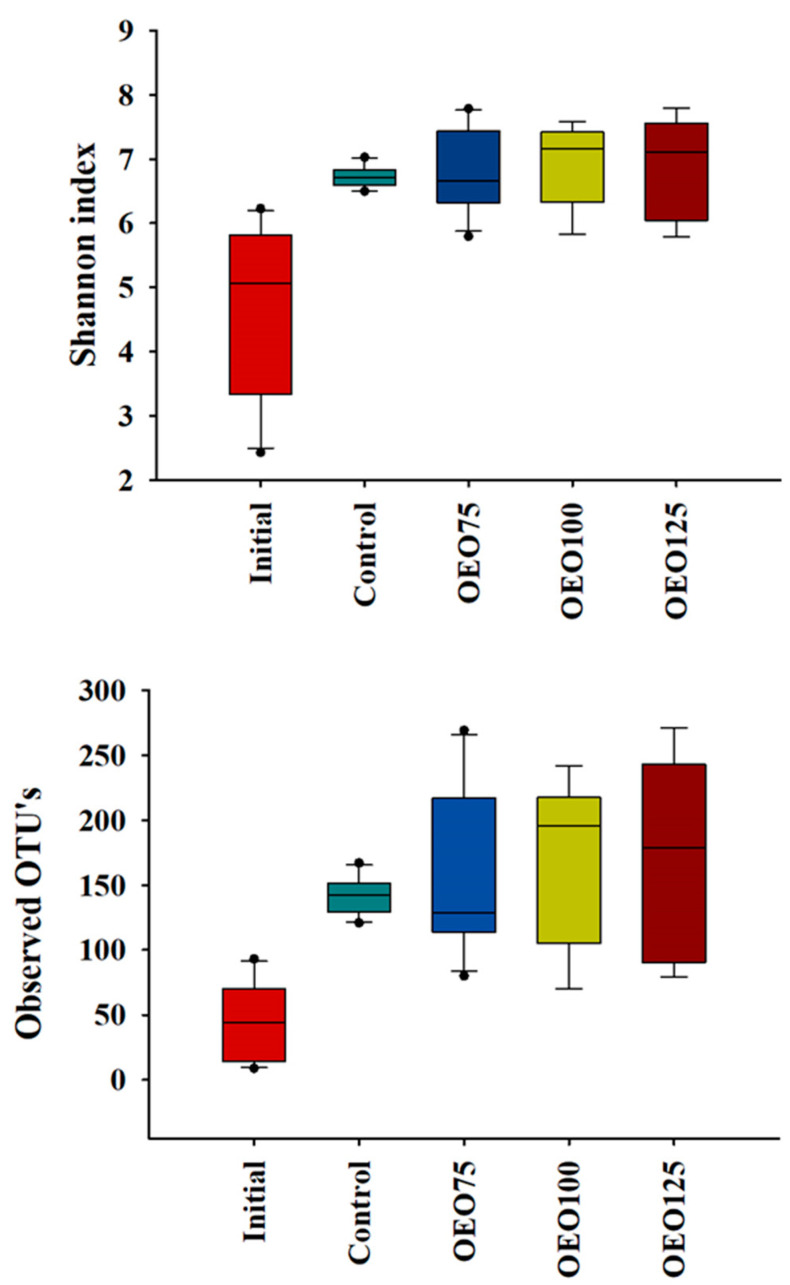
Box plot of Shannon Index and Observed OTUs for alpha diversity of the gut microbiota of striped bass experimental groups (Control, OEO75, OEO100, and OEO125) and initial samples. The control group exhibited less variability in the Shannon index and observed OTUs than the experimental groups. Additionally, initial samples show less richness and OTUs than the experimental groups.

**Figure 5 microorganisms-13-00264-f005:**
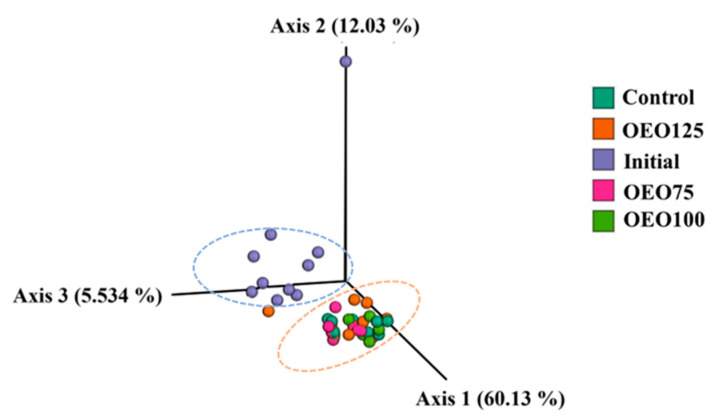
Principal coordinate analysis (PCoA) plot for beta diversity index of the gut microbiota of striped bass experimental groups (Control, OEO75, OEO100, and OEO125) and initial samples. The plot is based on the weighted phylogenetic distances of UniFrac.

**Figure 6 microorganisms-13-00264-f006:**
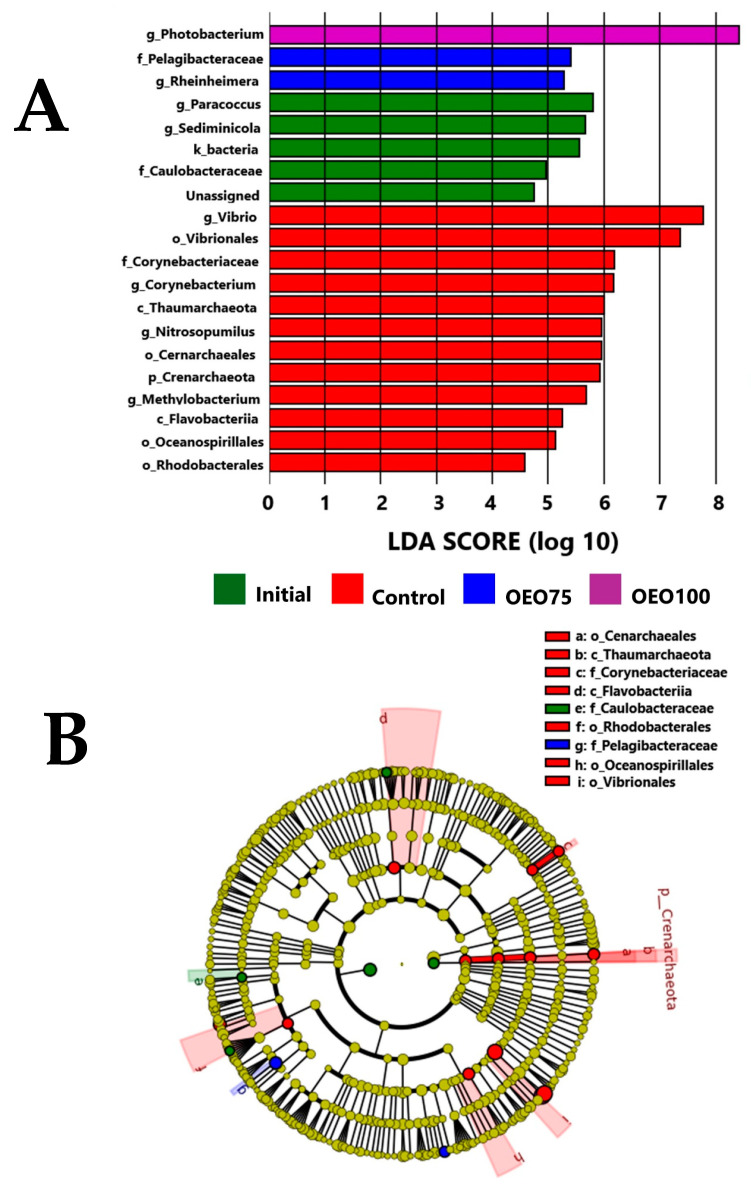
(**A**) Differential expressed taxa of the gut microbiota of striped bass experimental groups (Control, OEO75, and OEO100) and initial samples, using the LEFSe analysis and shown as log10 LDA score for each group. (**B**) The cladogram with the differences in the phylogenetic relation between the microbiota of each group is presented. The differences are presented in the labels in the right superior corner. Letters: p_(phyla), c_(class), o_(order), f_(family), and g_(genera).

**Figure 7 microorganisms-13-00264-f007:**
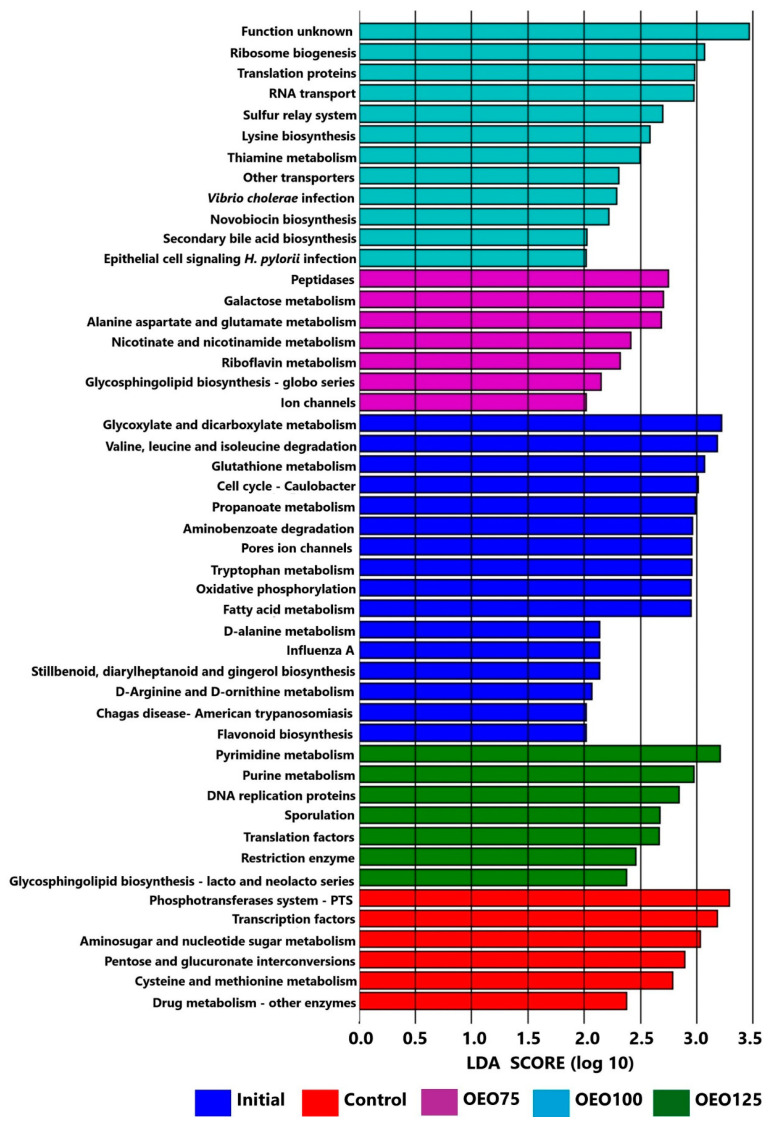
Enriched metabolic pathways in KEGG level 3 (class) predicted for the differentially expressed taxa in the gut microbiota of striped bass experimental groups (Control, OEO75, OEO100, and OEO125) and initial samples, using the LEFSe analysis and shown as log10 LDA score for each group. The image shows a few predicted functions (16 of 58) of the initial samples’ microbiota, which was the most enriched group. View the complete figure in the Appendix A.

**Table 1 microorganisms-13-00264-t001:** Growth performance and feed utilization parameters of juveniles of striped bass fed with OEO-supplemented diets.

Item	Oregano Essential Oil mL/kg			
	0	0.75	1.0	1.25
IBW (g)	109 ± 8.5	111.4 ± 8.5	107.2 ± 10.9	110.5 ± 9.3
FBW (g)	228.4 ± 38.4 ^a^	242.3 ± 24.2 ^b^	224.3 ± 36.8 ^a^	232.2 ± 36.9 ^ab^
WG (g)	119.20 ± 28.70	130.95 ± 7.46	117.10 ± 23.10	121.68 ± 24.26
WG (%)	109.12 ± 26.22	117.51 ± 6.48	108.58 ± 15.32	109.58 ± 17.65
SGR (%/day)	1.04 ± 0.17	1.11± 0.0	1.04 ± 0.1	1.05 ± 0.12
FCR	1.05 ± 0.0 ^a^	0.91 ± 0.0 ^b^	0.99 ± 0.0 ^ab^	1.00 ± 0.0 ^ab^
HSI	1.86 ± 0.1	1.78 ± 0.2	1.70 ± 0.2	1.67 ± 0.2
K_a_	2.48 ± 0.05	2.44 ± 0.02	2.46 ± 0.01	2.48 ± 0.04
Survival (%)	100 ± 0.0	100 ± 0.0	100 ± 0.0	100 ± 0.0

Means ± SD of the three replicates with different superscript letters indicate statistical differences (*p* < 0.05). IBW: Initial body weight; FBW: Final body weight; WG: Weight gain; SGR: Specific growth rate; FCR: Feed conversion ratio; HSI: Hepatosomatic index; K_a_: Allometric condition factor.

## Data Availability

The original raw sequence reads data is openly available in NCBI with the accession code PRJNA1149096 at https://www.ncbi.nlm.nih.gov/bioproject/1149096 (accesed on 16 August 2024).

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
