# Peer review of "Effect of Oregano Essential Oil on Growth and Composition of Gut Prokaryote Microbiota on Striped Bass (Morone saxatilis)"

_microorganisms, 2025, doi:10.3390/microorganisms13020264_

Round 1
Reviewer 1 Report
Comments and Suggestions for Authors
Manuscript entitle “Effect of oregano essential oil on growth and composition of gut prokaryote microbiota on striped bass (Morone saxatilis) ” investigated the effect of a diet supplemented with oregano oil on growth performance and gut bacterial and archaeal gut microbiota composition of juvenile striped bass (Morone saxatilis) raised in sea water in Ensenada, Baja California, México. Specifically, the addition of OEO didn’t have a direct effect on growth performance. However, it did reshape striped bass’s gut microbiota by reducing the relative and differential abundance of dominant opportunistic pathogens. These findings contribute knowledge on its possible inclusion as a sustainable dietary additive to optimize fish health and productivity.
Author Response
Comment: Manuscript entitle "Effect of oregano essential oil on growth and composition of gut prokaryote microbiota on striped bass (Morone saxatilis)" investigated the effect of a diet supplemented with oregano oil on growth performance and gut bacterial and archaeal gut microbiota composition of juvenile striped bass (Morone saxatilis) raised in sea water in Ensenada, Baja California, México. Specifically, the addition of OEO didn't have a direct effect on growth performance. However, it did reshape striped bass's gut microbiota by reducing the relative and differential abundance of dominant opportunistic pathogens. These findings contribute knowledge on its possible inclusion as a sustainable dietary additive to optimize fish health and productivity.
Response: We sincerely thank the reviewer for their thoughtful and comprehensive manuscript summary. We deeply appreciate your recognition of the importance of our findings. As researchers, it is our fundamental mission to generate knowledge that can improve aquaculture practices, especially given the growing significance of this sector for global food security and human development. Our study highlights the potential of oregano essential oil as a sustainable dietary additive to promote fish health by modulating gut microbiota and reducing opportunistic pathogens, thus contributing to more sustainable and productive aquaculture practices. We are committed to continuing our efforts to advance research that supports the development of innovative, sustainable solutions for aquaculture. Thank you again for your valuable feedback and encouragement.
Reviewer 2 Report
Comments and Suggestions for Authors
This manuscript (Microorganisms-3377223) under the title "Effect of oregano essential oil on growth and composition of gut prokaryote microbiota on striped bass (Morone saxatilis)" evaluated the influence of dietary oregano essential oil (OEO) supplementation on growth and microbiota composition in the intestine of striped bass. The diet containing OEO had no significant growth-promoting effect, but it showed improved performance in intestinal microbiome in striped bass. Results from this study confirmed this possibility of OEO to serve as a potential feed additive to modulate the gut microbiota and maintain gut heath in striped bass and provide the reference for developing safe, effective, and sustainable plant-derived feed additives in aquafeed.
This manuscript is valuable for the new dietary plant component in aquafeed and farmed aquatic animals. However, the presentation of abstract and introduction in this manuscript are less clear. The method part lacks of many critical information. Multiple unnecessary or over-long descriptions in the text, especially the result and discussion, would make readers more confused. Additionally, there are several mistakes in language, syntax, and format. Therefore, the authors need to reorganize and polish the whole manuscript to improve its quality.
Major comments:
1. In the "Abstract", critical information on this study is missing, such as the initial weight and total number of experimental fish, the final results of growth performance. Additionally, some statements are a bit abrupt in the main manuscript, and lacking in important details/context. For example, Line 16.
Thus, the current "Abstract" part should be clearly rephrased and provided more necessary information.
2. In the "1.Introduction" section, the relevant research information/background details is insufficient and unclear. For example, the brief background on the experimental fish, striped bass (Morone saxatilis), should be provided.
There are some really long complicated sentences. For example, Line 54-58, Line 75-80. These long statements may interfere with understanding. It is suggested to split the over-long and complicated sentences and add explanatory where applicable.
The cited references are missing from several descriptions. For example, the text in Line 63-65 and Line 68-69 lack of the citation of previous reports.
Additionally, there are some grammatical errors and non-standard English expressions, such as the presentation of volume unit and fish name.
Thus, the authors should rephrase the relevant statement of "1.Introduction" for a clear focus on background or significance in this study.
3. In the section of "2.Materials and Methods", multiple methodological descriptions on key information are missing.
First, prior to the feeding experiments, aquatic animals are generally cultured for one or two weeks to adapt to the laboratory conditions. Is there acclimatization in this study? How long? What feed were used during the acclimating period? Control diet or other feed?
Second, there was no information on the complete formulation and the proximate composition of the control diet and three experimental diets containing OEO.
Third, what about the aquatic environmental parameters (such as salinity, dissolved oxygen, ammonia nitrogen, etc,) during the feeding period? Were these parameters same as those in the acclimating period or not?
Fourth, use the correct presentation of volume unit and mass unit.
Thus, the authors should add more relevant information. Please revise "2.Materials and Methods" section accordingly.
4. In the "3.Results" part, there were several unnecessary and redundant descriptions. For example, in Line 196-197, Line 250-251, etc. The original text of Line 196-197 is a bit unnecessary for the core results. Because all the growth parameters of striped bass must be similar prior to the feeding trial. The sentence in Line 250-251 belongs to the method description. They could be deleted directly without any negative influence on the corresponding paragraph or moved to the method part.
Additionally, Table 1 lacks the legend. What is the meaning of superscript letter in Table 1? Please check and provide more information on the respective parameters in the legend of Table 1.
The presentation of results in this section should be streamlined with emphasis. Please modify accordingly.
5. The main content of "4.Discussion" section is not well-organized and has many grammatical errors and statement issues, such as lengthy and confusing sentences, descriptions where not relevant, missing cited references, grammatical and stylistic errors, etc.
For example, in Line 370-372, the references are missing in the text of discussion part. The statement in Line 388-391 is overly verbose and readers tended to be more confused by this lengthy description. It would be preferable to split into two short sentences. Similar errors regarding over-long statements are present in the other part of discussion.
Moreover, the text of Line 440-445 is more relevant to the main point of Line 446-456 (the 9th paragraph of discussion part) in the manuscript than that of Line 435-440. It would be preferable to move these sentences of Line 440-445 to the later paragraph (the 9th paragraph of discussion part) and incorporate the corresponding content.
Thus, the authors should reorganize the whole discussion section for better clarifying your findings and eliminating language errors.
Minor comments:
1. The current "Keywords" might not a good match to the main content of this paper. Please revise it by adding the correct term and removing the redundant one. For example, "oregano essential oil" should be included.
2. Please check the symbols for volume unit and mass unit in this study according to the information to related guides. For example, in Line 15-16, 74, 94, etc., please replace "ml kg-1" with "mL/kg", "g kg-1" with "g/kg". The authors need to check the whole page and modify accordingly.
3. The authors need to check the current reference list of this manuscript. The total number of reference could be reduced. When some references are cited to justify a fact, chose the most appropriate literatures and eliminate the remaining ones.
4. Page 20 contained the paper-irrelevant information. Please remove directly.
Other errors (highlighted in yellow) were marked in the PDF file.
So, this manuscript will be reconsidered after major revision.

Comments on the Quality of English Language
There are many lengthy descriptions in the main text of this paper. These complicated sentences made the reading difficult. The presentation in the main text, particularly the section of "Abstract" and "Introduction" contained insufficient descriptions. Also there are still several mistakes, such as the presenting for table and reference list. It is recommended that the text should be proofread by a professional or native speaker.
Author Response
|
3. Point-by-point response to Comments and Suggestions for Authors |
|
Comments 1: In the "Abstract", critical information on this study is missing, such as the initial weight and total number of experimental fish, the final results of growth performance. Additionally, some statements are a bit abrupt in the main manuscript, and lacking in important details/context. For example, Line 16. Thus, the current "Abstract" part should be clearly rephrased and provided more necessary information.
|
|
Response 1: Thank you for bringing this to our attention. We have updated the abstract by adding information about the initial weight, total number of fish, and growth performance results. Additionally, we have removed Line 16 and adequate the abstract to have more fluency.
Abstract: Sustainable strategies such as oregano essential oil (OEO) are being assessed to improve farmed fish's health and performance. Several studies in freshwater species describe the beneficial effects of OEO as a dietary supplement. Nonetheless, information about its use in marine fish is scarce. Hereby, this study consisted of four experimental groups: a control and three levels of OEO dietary inclusion: OEO75 (0.75 mL/kg), OEO100 (1.0 mL/kg), and OEO125 (1.25 mL/kg) with 23 fish of 110 ± 1.2 g per tank (n= 276) in a seawater flow-through system. After 70 days, data for growth parameters and samples for gut microbiota were taken. The final weight of OEO75 was higher (242.3 ± 24.2 g), and its feed conversion rate (0.91± 0.0) was reduced. However, these differences could be related to the sampling interval between the groups. Bioinformatic analysis of OEO groups revealed a reduction in Proteobacteria (Vibrio, Flavobacteria, and Vibrionales order) abundance and an increase in Euryarchaeota phyla in a dose-dependent manner. The predicted functions shifted from nutritional (OEO75) to replication, proliferation, and bacterial defense mechanisms (OEO100 and OEO125). These results show that adding OEO did not improve growth performance, but it reshaped the gut microbiota by reducing the abundance of dominant bacteria and modifying their metabolic pathways.
|
|
Comments 2: In the "1. Introduction" section, the relevant research information/background details is insufficient and unclear. For example, the brief background on the experimental fish, striped bass (Morone saxatilis), should be provided.
There are some really long complicated sentences. For example, Line 54-58, Line 75-80. These long statements may interfere with understanding. It is suggested to split the over-long and complicated sentences and add explanatory where applicable.
The cited references are missing from several descriptions. For example, the text in Line 63-65 and Line 68-69 lack of the citation of previous reports.
Additionally, there are some grammatical errors and non-standard English expressions, such as the presentation of volume unit and fish name.
Thus, the authors should rephrase the relevant statement of "1.Introduction" for a clear focus on background or significance in this study.
|
|
Response 2: We appreciate your comment and have incorporated a brief background of striped bass: Line 78-85: The striped bass (Morone saxatilis) is an anadromous, endemic fish of the North American Atlantic Coast and the Gulf of Mexico. Following their introduction to the West Coast, they are now distributed from British Columbia, Canada, to Baja California, Mexico[33]. Their potential for aquaculture is highlighted by their euryhaline nature, which allows them to live in marine, brackish, or freshwater. In addition, knowledge of their biological and rearing requirements is available due to the development of the hybrid striped bass (Morone chrysops × Morone saxatilis) culture [34]. Currently, Pacifico Aquaculture, located in Ensenada, México, is the only commercial farm that produces striped bass raised in sea cage culture.
Additionally, we have modified the long, complicated sentences in the introduction: Line 36-38: The aquaculture sector is exploring new sustainable strategies, such as phytogenics, to improve the growth and health of farmed fish by ensuring intestinal health and balance of gut microbiota [4,5].
Line 55-60: Recently, archaea have gained attention as potential probiotics due to their diverse beneficial qualities. Firstly, they can resist the routinary mechanical and thermic processes of animal feed production, thereby remaining viable to colonize the gut and prevent the invasion of pathogens [18,19]. To date, no pathogenic archaea have been identified. On the contrary, they are integral components of the microbiota associated with animals and humans, with the ability to modulate the host's immune system [19,20].
We have additionally included references to previously unreferenced statements where necessary: Line 65-67: Studies have documented the various properties of OEO, including its roles as an immu-noregulator, antioxidant, enhancer of gut morphology, stimulator of digestive enzymes, and modulator of microbiota [24-27]. Line 71-72: Recently, some studies on marine fish have reported the use and effects of OEO [3]
We have changed the presentation of volume unit and fish name in all the manuscript: Line 15: OEO75 (0.75 mL/kg), OEO100 (1.0 mL/kg), and OEO125 (1.25 mL/kg) Line 70-71: common carp (Cyprinus carpio) and Nile tilapia (Oreochromis niloticus) [25,28-31]. Line 72: In turbot (Scophthalmus maximus) [27], the administration of 1 ml/kg Line 76: an addition of 0.3 g/kg of OEO
|
|
Comments 3: In the section of "2.Materials and Methods", multiple methodological descriptions on key information are missing.
First, prior to the feeding experiments, aquatic animals are generally cultured for one or two weeks to adapt to the laboratory conditions. Is there acclimatization in this study? How long? What feed were used during the acclimating period? Control diet or other feed?
Second, there was no information on the complete formulation and the proximate composition of the control diet and three experimental diets containing OEO.
Third, what about the aquatic environmental parameters (such as salinity, dissolved oxygen, ammonia nitrogen, etc,) during the feeding period? Were these parameters same as those in the acclimating period or not?
Fourth, use the correct presentation of volume unit and mass unit.
Thus, the authors should add more relevant information. Please revise "2.Materials and Methods" section accordingly. |
|
Response 3: Thank you for pointing this out. We have added a paragraph addressing the acclimatization period and the feed used. Additionally, we incorporated the manufacturer's guaranteed analysis of the commercial diet as this feed was used during the acclimation period without the OEO, and later in the bioassay, the unique addition was the OEO via spraying.
In the paragraph below, we incorporate the environmental parameters recorded during the acclimation period.
Line 102-112: Striped bass (M. saxatilis) juveniles were donated by Pacifico Aquaculture SAP.I de C.V. located in Ensenada, Baja California, México. Before bioassay, fish had an acclimating period of one month in the experimental system, which consisted of a seawater flow-through system composed of 12 tanks of 430 L. Additionally, each tank had a temperature controller, a titanium heater of 1000 watts, and a temperature and water sensor. Water parameters were maintained as follows: water temperature (23.0 ± 0.04 °C), salinity (33.9 ± 0.5 ‰), dissolved oxygen (6.5 ± 0.3 mg/L), TAN (< 0.5 mg/L ), nitrite (< 0.25 mg/L) and nitrate (< 5.0 mg/L). During this period, animals were fed with the commercial diet EWOS® (Vancouver, BA, Canada) comprising 54 % crude protein, 14 % crude fat, 10 % nitro-gen-free extractives, 0.6 % crude fiber, and 11 % ash, according to the manufacturer's guaranteed analysis.
We have also corrected the presentation of the volume unit and mass unit: Line 115: were in each tank with a water flow of 1.50 L/min Line 117: oxygen (6.5 ± 0.07 mg/L) were recorded daily Line 118: While TAN (≤ 0.5 mg/L), nitrite (< 0.25 mg/L), and nitrate (< 5.0 mg/L) were Line 133-135: a control group (0 mL/kg) and three treatments with OEO: OEO75 (0.75 mL/kg ) OEO100 (1.0 mL/kg), and OEO125 (1.25 mL/kg). Line 151: using 0.05 mL/L clove oil
|
|
Comments 4: In the "3.Results" part, there were several unnecessary and redundant descriptions. For example, in Line 196-197, Line 250-251, etc. The original text of Line 196-197 is a bit unnecessary for the core results. Because all the growth parameters of striped bass must be similar prior to the feeding trial. The sentence in Line 250-251 belongs to the method description. They could be deleted directly without any negative influence on the corresponding paragraph or moved to the method part.
Additionally, Table 1 lacks the legend. What is the meaning of superscript letter in Table 1? Please check and provide more information on the respective parameters in the legend of Table
The presentation of results in this section should be streamlined with emphasis. Please modify accordingly. |
|
Response 4: Thank you for pointing this out. We have removed Lines 196-197, 250-251, and 257.
In addition, we have added the proper legend to the Table 1: Below Table 1: Means ± SD of the three replicates with different superscript letters indicate statistical differences (p < 0.05). IBW: Initial body weight; FBW: Final body weight; WG: Weight gain; SGR: Specific growth rate; FCR: Feed conversion ratio; HSI: Hepatosomatic index; Ka: Allometric condition factor.
|
|
Comments 5: The main content of "4.Discussion" section is not well-organized and has many grammatical errors and statement issues, such as lengthy and confusing sentences, descriptions where not relevant, missing cited references, grammatical and stylistic errors, etc.
For example, in Line 370-372, the references are missing in the text of discussion part. The statement in Line 388-391 is overly verbose and readers tended to be more confused by this lengthy description. It would be preferable to split into two short sentences. Similar errors regarding over-long statements are present in the other part of discussion.
Moreover, the text of Line 440-445 is more relevant to the main point of Line 446-456 (the 9th paragraph of discussion part) in the manuscript than that of Line 435-440. It would be preferable to move these sentences of Line 440-445 to the later paragraph (the 9th paragraph of discussion part) and incorporate the corresponding content.
Thus, the authors should reorganize the whole discussion section for better clarifying your findings and eliminating language errors |
|
Response 5: We appreciate your coment. We have modified the discussion accordingly.
We added the references needed in the following statements: Line 407-409: Several studies have documented the effects of OEO as a growth enhancer by boosting final weight, weight gain, feed conversion ratio, condition index, and protein efficiency ratio [3,31,50,51].
We modified the long descriptions, trimming them in two sentences: Line 426-428: This result agrees with the reported in channel fish (Ictalurus punctatus) fed with diets containing the same OEO product used in our study, these fish had a lower HIS. On the contrary, the fish that received other experimental diets (Thymol, Carvacrol, and Carvacrol + Thymol) showed a higher HSI [51].
Line 463-466: Numerous studies have demonstrated the antibacterial properties of OEO, highlighting its notable lipophilic characteristics. This property of OEO damages the integrity of the cell membrane, altering its stability and causing cell death [5, 62-64].
Line 515-520: Studies in channel catfish, koi carp, rainbow trout, and gilthead seabream (Sparus aurata L.) have reported the immunomodulation of oregano by improving innate immunity, enhancing the activity of lysozyme, complement activity, and levels of immunoglobulin M. Additionally, promoting cellular immunity by increasing lymphocytes number and phagocytic activity of leucocytes [26,51,81,82].
We incorporated the indicated sentences to the 9th paragraph of the discussion. The final version of the paragraph is shown below: Line 478-482: Archaea, although typically less abundant than bacterial communities in fish gut ecosystems, possess unique adaptations such as cell wall structures that enable survival in extreme conditions, along with distinctive metabolic processes like methanogenesis [68,69]. Methanogens from Euryarchaeota play a crucial role in the final stages of carbohydrate metabolism, converting organic matter by-products into methane [69,70]. On the other hand, the antibacterial effects of OEO have been predominantly observed against Gram-positive and Gram-negative bacteria, with limited information on its impact on archaea [62,71].
|
|
Comments 6: The current "Keywords" might not a good match to the main content of this paper. Please revise it by adding the correct term and removing the redundant one. For example, "oregano essential oil" should be included. |
|
Response 6: We appreciate your comment. We have incorporated "oregano essential oil" and removed "phytogenics". The final keywords are shown below:
Keywords: 16s rRNA, amplicon sequencing, oregano essential oil, striped bass, prokaryote gut microbiota.
|
|
Comments 7: Please check the symbols for volume unit and mass unit in this study according to the information to related guides. For example, in Line 15-16, 74, 94, etc., please replace "ml kg-1" with "mL/kg", "g kg-1" with "g/kg". The authors need to check the whole page and modify accordingly. |
|
Response 7: Thank you for your comment. We modified the symbols for volume unit and mass unit in all the manuscript. You can find the modified lines in the response to your comment numer 2 and 3.
|
|
Comments 8: The authors need to check the current reference list of this manuscript. The total number of reference could be reduced. When some references are cited to justify a fact, chose the most appropriate literatures and eliminate the remaining ones. |
|
Response 8: We agree with your comment. We have reduced the reference list from 92 to 87, as we had to add four new references for the 4th paragraph of the introduction and the 14th paragraph of the discussion.
|
|
Comments 9: Page 20 contained the paper-irrelevant information. Please remove directly. |
|
Response 9: Thank you for pointing this out. We have removed the page 20.
|
|
4. Response to Comments on the Quality of English Language |
|
Point 1: Additionally, there are several mistakes in language, syntax, and format. Response to English Quality: The English language in the manuscript has been thoroughly reviewed by all authors and refined using English editing software, such as Grammarly. Additionally, Dr. Cruz-Flores, a native English speaker, conducted an extensive revision of the manuscript. |
|
|
|
5. Additional clarifications |
|
All queries raised by the reviewers have been addressed. |
Reviewer 3 Report
Comments and Suggestions for Authors
microorganisms-3377223-peer-review-v1
The paper presents an interesting research approach for the application of essential oils in the improvement of health of striped bass, where experiments carried out with controls, and 3 different levels of essential oil were investigated. In addition, authors have performed metagenomic analysis to investigate changes in the microbial GIT environment and further analyzed and suggested benefits form the modulation of the microbiota for the health properties of the fiches.
In the preparation of the manuscript, authors provided reasonable introduction where justification of application of essential oils were stated and pointed of need of new approaches in the development of sustainable aquaculture. Further, material and methods are described with sufficient details. An interesting point is that authors have performed the metagenomic analysis based on DNA isolated from the experimental fiches. It is a known fact that DNA can give some results, that will refer to the DNA form already dead cells, and do not really represent status of the life microbial population. How authors will comment on this issue, are some modifications, some additional experiments adjustments applied with aim to not misinterpret the obtained results? Please, can this issue be pointed out and discussed in the paper?
Further, results are represented with sufficient details and illustrative materials. Maybe authors can reconsider moving some of the illustrative material as supplementary?
Discussion is well structured and presented. In my opinion authors have covered essential points of the subject and constructed an interesting paper.
The problem can be fact that 39% similarity were detected. Will be appropriate if authors can revise the manuscript and reduce this similarity levels to acceptable percentage. The paper will need to be corrected as well regarding language issues. Maybe help from more experienced colleagues or language professional will be appropriate choice regarding this issue.
Ln86: Please, provide the address for the mentioned research center. The experiments are conducted according to the recommendations from the agencies, however, authors have a ethical commission authorization for the performing these experiments? Mexican regulations require such as authorization?
Please check the references, some of them are not in recommended format.
All material and equipment mentioned in the manuscript needs to be accompanied by appropriate information regarding suppliers. Generally, for each material address of the supplier needs to be provided including name of the company, city, state (in case of federal country, where state name needs to be provided in abbreviated way) and name of the country. In second and following occasion, only name of the company will be sufficient.
Author Response
|
3. Point-by-point response to Comments and Suggestions for Authors |
|
Comments 1: An interesting point is that authors have performed the metagenomic analysis based on DNA isolated from the experimental fiches. It is a known fact that DNA can give some results, that will refer to the DNA form already dead cells, and do not really represent status of the life microbial population. How authors will comment on this issue, are some modifications, some additional experiments adjustments applied with aim to not misinterpret the obtained results? Please, can this issue be pointed out and discussed in the paper?
|
|
Response 1: We agree with your comment, and we added a paragraph at the end of the discussion addressing the limitations of the technique and the additional ways to overcome this issue.
Line 537-549: The 16S rRNA amplicon sequencing technology allowed us to identify the archaeal and bacterial taxa of striped bass's gut microbiota and its variations between the experimental groups. However, it is important to acknowledge that a limitation of 16S rRNA amplicon sequencing analysis is its inability to distinguish between environmental DNA and DNA from viable cells. Unlike most RNA species, which quickly degrades, DNA remains detectable after a cell's death [86]. Therefore, a strategy to determine living taxa is using RNA to obtain complementary DNA and then applying the 16S rRNA amplicon sequencing analysis [87]. However, cost and differences in processing steps should be considered. Although the RNA approach would have increased analysis sensitivity, the acquired data allowed for comprehensive analysis. These findings are valuable for innovating and applying new sustainable alternatives based on aromatic plants like oregano.
|
|
Comments 2: Further, results are represented with sufficient details and illustrative materials. Maybe authors can reconsider moving some of the illustrative material as supplementary? |
|
Response 2: Thank you for your valuable suggestion. We agree that illustrative materials play a crucial role in enhancing the clarity and impact of our findings. However, we believe that retaining the figures within the main manuscript is essential to ensure that each reported result is presented comprehensively and with the necessary context. This approach helps readers easily follow and understand the study's findings without the need to navigate supplementary materials. Nevertheless, if the editorial team deems it necessary, we are open to relocating some figures to the supplementary section while ensuring the manuscript remains cohesive and reader-friendly.
|
|
Comments 3: The problem can be fact that 39% similarity were detected. Will be appropriate if authors can revise the manuscript and reduce this similarity levels to acceptable percentage. |
|
Response 3: Thank you for highlighting this concern. We have thoroughly revised the manuscript and re-evaluated its similarity using the Grammarly extension for Word, which now indicates a reduced similarity level of 15%. Additionally, we would like to emphasize that the majority of the detected similarity originates from the reference list, rather than the core sections of the manuscript, such as the introduction, materials and methods, results, discussion, or conclusion. We believe this revised percentage aligns with acceptable standards, and we hope it addresses your concern satisfactorily.
|
|
Comments 4: Ln86: Please, provide the address for the mentioned research center.
|
|
Response 4: Thank you for your comment. We added the location of the research center, as the full address of the research center is below the authors' names. Line 96: located in Ensenada, Baja California, México.
Line 6: 1 Centro de Investigación Científica y Educación Superior de Ensenada, (CICESE) Carretera Ensena-da-Tijuana No. 3918, Zona Playitas, 22860. Ensenada, Baja California, México
|
|
Comments 5: The experiments are conducted according to the recommendations from the agencies, however, authors have a ethical commission authorization for the performing these experiments? Mexican regulations require such as authorization?
|
|
Response 5: Thank you for raising this important concern regarding ethical authorization. In Mexico, current regulations and norms do not yet mandate ethical authorization for certain types of scientific research, including the experiments described in our study. Nevertheless, we recognize the importance of adhering to ethical standards. Prior to submitting this manuscript, we initiated the process of obtaining approval from the ethical commission of our research center by submitting the required documentation. Unfortunately, due to holiday-related delays, the review process has not yet been finalized. We anticipate receiving the corresponding approved response from the ethical commission as soon as the administrative operations resume, and we are committed to updating the editorial team promptly with the official documentation once it is available. We appreciate your understanding and commitment to ethical research practices.
|
|
Comments 6: Please check the references, some of them are not in recommended format. |
|
Response 6: Thank you for pointing this out. We have checked and updated the references according to the correct format. You can find the modifications on the references highlighted in the manuscript.
|
|
Comments 7: All material and equipment mentioned in the manuscript needs to be accompanied by appropriate information regarding suppliers. Generally, for each material address of the supplier needs to be provided including name of the company, city, state (in case of federal country, where state name needs to be provided in abbreviated way) and name of the country. In second and following occasion, only name of the company will be sufficient. |
|
Response 7: Thank you for your comment. We added the appropriate information regarding suppliers in the lines below. Line 110: EWOS® (Cargill, Vancouver, BA, Canada) Line 117: a multiparameter (YSI, Yellow Spring, OH, USA) Line 119: with a commercial kit (API®, Chalfond, PA, USA) Line 122: Anpario plc. (Nottinghamshire, United Kingdom) Line 124: EWOS® (Cargill). Line 126: the diet was mixed in a blender (Member's Mark, Bentonville, AR, USA) Line 128: incubator (Boekel Scientific, Philadelphia, PA, USA) Line 152: clove oil (Sigma-Aldrich, St. Louis, MO, USA) Line 155: ultra-freezer (Thermo Fisher Scientific, Waltham, MA, USA) Line 157: DNAeasy® Blood & Tissue (Qiagen, Hilden, Germany) Line 159: Spectrophotometer (Thermo Fisher Scientific). Line 162: QuantiFluor® dsDNA System (Promega, Madison, WI, USA) Line 164: KAPA HyperPlus (Roche, Basel, Switzerland) Line 167: illumina® MiSeq platform (Illumina, Inc., San Diego, CA, USA) Line 208: Statistica version 10.0 software (StatSoft In., Tulsa, OK, USA).
|
|
4. Response to Comments on the Quality of English Language |
|
Point 1: The paper will need to be corrected as well regarding language issues. Maybe help from more experienced colleagues or language professional will be appropriate choice regarding this issue. |
|
Response to English Quality: The English language in the manuscript has been thoroughly reviewed by all authors and refined using English editing software, such as Grammarly. Additionally, Dr. Cruz-Flores, a native English speaker, conducted an extensive revision of the manuscript. |
|
5. Additional clarifications |
|
All queries raised by the reviewers have been addressed. |
Round 2
Reviewer 2 Report
Comments and Suggestions for Authors
The revised manuscript (Microorganisms-3377223) under the title "Effect of oregano essential oil on growth and composition of gut prokaryote microbiota on striped bass (Morone saxatilis)" has been adequately modified as suggested. Moreover, the authors reply to the reviewer’s comments one by one and explain how they revised the manuscript (in cover letter).
The current revision is suitable for publication in your journal, though the reference list still contains some minor errors, such as volume and page number. For example, in Reference 37, the correct information is 2018, volume 10, issue 3, page 716-726.